# Transcriptome Analysis Reveals Critical Genes and Pathways in Carbon Metabolism and Ribosome Biogenesis in Poplar Fertilized with Glutamine

**DOI:** 10.3390/ijms23179998

**Published:** 2022-09-02

**Authors:** Mei Han, Mingyue Xu, Tao Su, Shizhen Wang, Liangdan Wu, Junhu Feng, Changjun Ding

**Affiliations:** 1Co-Innovation Center for Sustainable Forestry in Southern China, College of Biology and the Environment, Nanjing Forestry University, Nanjing 210037, China; 2Key Laboratory of State Forestry Administration on Subtropical Forest Biodiversity Conservation, Nanjing Forestry University, Nanjing 210037, China; 3Key Laboratory of Tree Breeding and Cultivation of State Forestry Administration, State Key Laboratory of Tree Genetics and Breeding, Research Institute of Forestry, Chinese Academy of Forestry, Beijing 100091, China

**Keywords:** transcriptome, *Populus*, L-glutamine, photosynthesis, sugar metabolism

## Abstract

Exogenous Gln as a single N source has been shown to exert similar roles to the inorganic N in poplar ‘Nanlin895′ in terms of growth performance, yet the underlying molecular mechanism remains unclear. Herein, transcriptome analyses of both shoots (L) and roots (R) of poplar ‘Nanlin895’ fertilized with Gln (G) or the inorganic N (control, C) were performed. Compared with the control, 3109 differentially expressed genes (DEGs) and 5071 DEGs were detected in the GL and GR libraries, respectively. In the shoots, Gln treatment resulted in downregulation of a large number of ribosomal genes but significant induction of many starch and sucrose metabolism genes, demonstrating that poplars tend to distribute more energy to sugar metabolism rather than ribosome biosynthesis when fertilized with Gln-N. By contrast, in the roots, most of the DEGs were annotated to carbon metabolism, glycolysis/gluconeogenesis and phenylpropanoid biosynthesis, suggesting that apart from N metabolism, exogenous Gln has an important role in regulating the redistribution of carbon resources and secondary metabolites. Therefore, it can be proposed that the promotion impact of Gln on poplar growth and photosynthesis may result from the improvement of both carbon and N allocation, accompanied by an efficient energy switch for growth and stress responses.

## 1. Introduction

Plant growth is determined by different kinds of environmental factors, such as water and nutrients [1]. Among nutrients, nitrogen (N) is one of the main elements limiting plant growth and development. N participates in many important metabolic processes of substances, therefore greatly influencing plant biomass accumulation and stress defenses [2,3]. In general, an inadequate supply of N often results in growth retardant, yield loss and inefficient water use efficiency [4], while adequate N positively affects plant physiology and biomass production in various crops [5,6]. Poplar, a fast-growing woody plant with high production and substantial carbon allocation to stem, is one of China’s main afforestation tree species. Its growth is mainly dependent on N [7,8]. Previous transcriptome analysis reveals that the regulatory effect of N fertilizer on poplar growth and development predominantly takes place at the transcriptional level, as apparent differences between gene expression patterns in response to adequate and surfeit inorganic N in poplar have been documented [9,10]. Overall, an increased N supply induces the expression level of organic amino acids metabolism-related genes, such as *gluco-6-phosphate dehydrogenase*, *phosphoenolpyruvate carboxylase*, *phosphoenolpyruvate carboxykinase*, *alanine aminotransferase* and *NADP-malic enzyme*, as a result of an increase demand for N assimilation and amino acids synthesis, and more carbon flowing to the synthesis of organic acids [9]. In addition, promoted expression of *hexokinase* and *sucrose synthase* was observed in poplar supplemented with high N, indicating that sucrose metabolism plays a vital role in carbohydrates production and sugar delivery from photosynthetic source organs to other tissues under high N conditions [9,10].

Although NO_3_^−^ and NH_4_^+^ have long been considered the primary N sources absorbed by higher plants [11], organic N in the form of amino acids represents an important proportion of exchangeable and soluble N pools in agricultural soils [12]. They have been demonstrated to be preferentially accessed by plants under certain conditions (such as limitation of inorganic N) [13,14]; meanwhile, a growing number of amino acid transporters have been identified in plants. These findings indicate that plants might take up and utilize amino acids’ form of N in a larger quantity and more prevalently than has been estimated [15,16,17].

Amino acids are components of proteins participating in many metabolic processes of organisms and play an essential role in plant nutrition [18]. Compared with the conventional inorganic N fertilizer, amino acids’ form of N have many advantages, for instance, maintenance of pH homeostasis in both plants and soil, since neither overproduction of proton nor overuse of proton occurred when amino acids were taken up as the dominant N source [19,20]. In addition, amino acids have great abilities to chelate metals [20] and to improve the quality, biomass [21,22] and N use efficiency of plants [23].

Glutamine (Gln), a proteinogenic amino acid that can be used as an N transporter and NH_4_^+^ carrier, participates in various molecular synthesis pathways, providing a source of energy, carbon and N skeleton to maintain biomass accumulation and cell homeostasis in living organisms [24,25]. Gln plays fundamental roles in different basic biological processes, including chlorophyll synthesis and TCA metabolism in plants [26]. For example, Gln with an optimal concentration increases chlorophyll content, promotes rice growth [27] and stimulates the vegetative growth of onion by retarding the destruction of chlorophyll or increasing the biosynthesis of chlorophyll and/or stabilizing the thylakoid membrane [28]. Additionally, the exogenous application of Gln increased plant height, leaf area and leaf SPAD value of basil plants, which has been supposed to be a consequence of the optimization of meristematic cells functions that enhance cell division and enlargement [29,30]. In addition, Gln has been considered an indispensable compound to maintain cell integrity and function in the biosynthetic pathway of nucleotides, amino sugars and nicotinamide adenine dinucleotide phosphate (NADPH) [24,31].

Our previous study using Gln as a sole N source to explore the biological effects of Gln on poplar ‘Nanlin895’ has demonstrated that Gln (0.5 mM) positively impacts poplar physiological parameters and phenotype. Exogenous application of Gln significantly improved the biomass production, photosynthetic rate and N use efficiency (NUE) of poplar compared with the N-free treatment, and it even displays a resemblance growth effect to the inorganic N (i.e., NO_3_^−^ plus NH_4_^+^) control [25]. Although these research works have demonstrated that Gln-based N significantly affects plant growth and development, the molecular mechanism by which such responses emerge has not been satisfactorily clarified. In this study, the transcriptional profiling of poplars fertilized with 0.5 mM Gln (G) and 2 mM inorganic N (control, C) was implemented, and the Gene Ontology (GO) and Kyoto Encyclopedia of Genes and Genomes (KEGG) functional annotations were used to analyze the biological pathways. In addition, quantitative real-time polymerase chain reaction (qRT-PCR) was implemented to validate the reliability of the transcriptome data.

## 2. Results

### 2.1. Transcriptome Data Analysis and Quality Evaluation of the cDNA Libraries

To investigate the molecular regulation mechanism of exogenous Gln on poplar growth, RNA-Seq was performed on the shoots (L) and roots (R) (three biological replicates in each group) of the inorganic N (control, C) and 0.5 mM Gln (G) fertilized poplar ‘Nanlin895’, respectively. In total, both CL and GL generated 44 million raw reads, while both CR and GR generated 43 million raw reads. After removing low-quality reads and adaptors, a total of 43 million and 41 million clean reads were generated from both control (CL and CR) and the Gln (GL and GR) libraries, respectively (Table 1). In all samples, the Q30 value was over 93%, and the guanine and cytosine content was above 43%. Over 87% of clean reads in the CL and GL could be aligned to the genome of *Populus trichocarpa*, of which 85% reads could only be uniquely aligned to one location on the genome. By contrast, over 85% and 74% of clean reads in the CR and GR could be aligned to the genome, of which 83% and 72% of reads could only be uniquely aligned to one location on the genome, respectively (Table 1). To confirm the reliability and rationality of the experiment, we calculated the Pearson’s correlation coefficient for all gene expression levels between each sample and displayed these coefficients in the form of a correlation matrix map (Figure 1a). In addition, the principal component analysis (PCA) was used to compare all of the samples (Figure 1b), from which a high similarity was found in the biological replicates. The above results revealed that the RNA-Seq data of these four cDNA libraries were accurate and reliable and could be used for subsequent analysis.

### 2.2. DEGs Obtained under Gln Treatment

The differentially expressed genes (DEGs) were measured based on the RNA-Seq reads abundance and were normalized to fragments per kilobase length per million reads (FPKM). A total of 3129 (including 1234 upregulated genes and 1895 downregulated genes) and 5071 (including 2824 upregulated genes and 2247 downregulated genes) DEGs among GL vs. CL and GR vs. CR (Figure 2a) were detected, respectively. A Venn diagram shows the number of identified DEGs, in which 910 DEGs overlap between the GL vs. CL and GR vs. CR (Figure 2b).

### 2.3. DEGs Classification and GO Functional Annotation

To investigate the potential biological functions of the DEGs between the GL vs. CL and GR vs. CR, we performed Gene Ontology (GO) analysis. The GO terms were divided into three major functional categories: biological process (BP), cellular component (CC) and molecular function (MF). Among all of the GO terms, the first 10 items in each functional category demonstrated that the most significant enrichments were presented (Figure 3). Regarding GL vs. CL, DEGs were mainly categorized into cellular carbohydrate metabolic process (GO:0044262), cytosolic ribosome (GO:0022626) and structural constituent of ribosome (GO:0003735) (Figure 3a), while DEGs between GR vs. CR were mainly enriched in the antibiotic metabolic process (GO:0016999), cell wall (GO:0005618) and peroxidase activity (GO:0004601) (Figure 3b). Furthermore, in the BP classification, a total of 92 DEGs and 98 DEGs were enriched in the cellular carbohydrate metabolic process (GO:0044262, 44 genes upregulation and 48 genes downregulation) and antibiotic metabolic process (GO:0016999, 85 genes upregulation and 13 genes downregulation) among GL vs. CL and GR vs. CR, respectively. In the CC classification, a total of 115 DEGs and 99 DEGs were enriched in the cytosolic ribosome (GO:0022626, 2 genes upregulation and 113 downregulation) and cell wall (GO:0005618, 58 genes upregulation and 41 downregulation) among GL vs. CL and GR vs. CR, respectively. In the MF classification, a total of 140 DEGs and 60 DEGs were enriched in the structural constituent of ribosome (GO:0003735, three genes upregulation and 137 genes downregulation) and peroxidase activity (GO:0004601, 51 genes upregulation and nine genes downregulation) among GL vs. CL and GR vs. CR, respectively (Figure 4).

The KEGG (Kyoto Encyclopedia of Genes and Genomes) functional classification of DEGs in GL vs. CL and GR vs. CR was performed to discover the metabolic pathways genes involved in Gln treatment. The 20 common metabolic pathways with the most significant enrichment in GL and GR by comparison with CL and CR, respectively, were exhibited. Among the top pathways in the GL vs. CL, a great number of DEGs participating in the pathway of the ribosome (pop03010), starch and sucrose metabolism (pop00500), and galactose metabolism (pop00052) (Figure 5a) were detected. On the contrary, the most enriched KEGG pathway of GR vs. CR was found in carbon metabolism (pop01200), phenylpropanoid biosynthesis (pop00940), biosynthesis of amino acids (pop01230) and glycolysis/gluconeogenesis (pop00010) (Figure 5b). Noteworthily, plant hormone signal transduction (pop04075) was enriched in both GL vs. CL and GR vs. CR.

### 2.4. DEGs Involved in Ribosome

In the process of plant growth and development, the formation of new cells and almost all the metabolic pathway needs the participation of protein. As a specialized multimolecular particle, the ribosome is widely involved in protein biosynthesis in all living cells. The 80S ribosome in eukaryotes is composed of a large subunit and small subunit, and their biosynthesis involves the production and correct assembly of four kinds of ribosomal RNA (rRNA), including 5.8S, 28S, 18S and 5S, and approximately 80 kinds of ribosomal protein (RP). In our study, a total of 140 DEGs in the ribosome pathway, including 40S ribosomal protein, 60S ribosomal protein, RP-related genes (i.e., 60S acidic ribosomal protein, ubiquitin-40S ribosomal protein and ubiquitin-60S ribosomal protein), 50S ribosomal protein and 30S ribosomal protein were identified by the pairwise comparisons of GL versus CL. A heatmap was generated based on the log_10_ (FPKM + 1) values of differentially expressed ribosome genes (Figure 6a). Except for three upregulated genes (*POPTR_001G365600*, *POPTR_001G262400*, *POPTR_013G070100*) (Figure 6b) located in the chloroplasts, all the ribosome DEGs were downregulated in the GL libraries.

### 2.5. Sugar and Carbon Metabolism in Response to Gln Treatment

Sugar metabolism is extremely vital for poplar growth and development. In the current study, a total of 42 DEGs (including 23 significant upregulated genes and 19 downregulated genes) in the GL libraries, and 46 DEGs (including 30 upregulated genes and 16 downregulated genes) in the GR libraries (Table 2) were annotated in starch and sucrose metabolism. Among sucrose biosynthesis genes, the expression of two sucrose synthases (*POPTR_004G081300* and *POPTR_002G202300*) and two sucrose phosphate synthases (*POPTR_018G124700* and *POPTR_013G095500*), two starch synthase (*POPTR_015G012400* and *POPTR_004G014100*) and two 1,4-alpha-glucan-branching enzyme genes (*POPTR_005G251000* and *POPTR_006G115100*) involved in amylose synthesis, and one beta-amylase (*POPTR_010G062900*) involved in the decomposition of amylose were upregulated in the GL compared with the CL. Additionally, three genes encoding hexokinase (*POPTR_018G088300*, *POPTR_009G050000* and *POPTR_001G254800*) and three genes encoding beta-glucosidase (*POPTR_019G037400*, *POPTR_001G403900* and *POPTR_008G094200*), five genes encoding endoglucanase (*POPTR_001G092200*, *POPTR_001G097900*, *POPTR_003G151700*, *POPTR_T144300* and *POPTR_015G126900*), four genes encoding probable trehalose phosphate phosphatase (*POPTR_007G090900*, *POPTR_005G077200*, *POPTR_012G126100* and *POPTR_014G126900*) and three genes encoding trehalose phosphate synthase (*POPTR_011G103900*, *POPTR_010G104500* and *POPTR_011G070900*) were identified. In addition, the transcripts of genes encoding alpha-amylase 3 (*POPTR_002G014300*), phosphoglucomutase (*POPTR_010G109500*) and glucose-6-phosphate isomerase 1 (*POPTR_008G118900*) were increased in the GL libraries (Table 2). Heatmap enrichment of the most significantly changed genes in sugar metabolism was shown in Figure 7a.

A large number of DEGs, including 92 upregulated genes and 35 downregulated genes enriched in carbon metabolism in the GR libraries, were annotated. Among them, genes encoding triosephosphate isomerase (*POPTR_008G056300*), D-3-phosphoglycerate dehydrogenase (*POPTR_009G096600*), phosphofructokinase 3/7 (*POPTR_006G235100* and *POPTR_018G069200*), aspartate aminotransferase 1/2 (*POPTR_006G107100* and *POPTR_018G082500*), malate dehydrogenase (*POPTR_004G112800*), pyruvate kinase 1 (*POPTR_010G233200* and *POPTR_008G002500*), enolase (*POPTR_015G131100*), glyceraldehyde-3-phosphate dehydrogenase C2 (*POPTR_010G055400*) and phosphoglycerate mutase (*POPTR_016G142900*) were significantly upregulated, while genes encoding 3-phosphoglycerate dehydrogenase (*POPTR_014G022800*) was strongly downregulated (Figure 7b).

Galactose is the precursor of the synthesis of raffinose and galactitol, and the accumulation of raffinose plays an important role in the process of carbon redistribution. In this study, the exogenous addition of Gln resulted in 24 DEGs in GL compared with CL, and 88 DEGs in phenylpropanoid biosynthesis in GR versus CR. Seven genes were annotated as galactinol synthase, among which six genes (*POPTR_014G116800*, *POPTR_008G101000*, *POPTR_010G150400*, *POPTR_010G042000*, *POPTR_013G005900* and *POPTR_013G005800*) were upregulated and one gene (*POPTR_002G191600*) was downregulated. Five genes were annotated as raffinose synthase, among which three genes (*POPTR_016G054700*, *POPTR_006G052800* and *POPTR_017G036700*) were upregulated and two genes (*POPTR_011G166700* and *POPTR_006G065700*) were downregulated. Additionally, one upregulated gene (*POPTR_003G037000*) and one downregulated gene (*POPTR_001G027400*) were annotated as beta-galactosidase, two downregulated (*POPTR_009G050000* and *POPTR_001G254800*) and one upregulated (*POPTR_018G088300*) genes were annotated as hexokinase, and two upregulated genes (*POPTR_018G152200* and *POPTR_010G255200*) were annotated as alpha-galactosidase. Genes encoding stachyose synthase (*POPTR_014G118400*) and phosphoglucomutase (*POPTR_010G109500*) were downregulated. In addition, two genes (*POPTR_015G127100* and *POPTR_003G112600*) annotated as acid beta-fructofuranosidase were downregulated (Table 3). The most significantly changed genes relevant to galactose metabolism are shown in Figure 8a.

Glycolysis and gluconeogenesis are two processes involved in the metabolism of glucose, which is the energy source of almost all the life forms on earth. In our study, 71 DEGs, including 62 upregulated genes and 9 downregulated genes annotated in glycolysis/gluconeogenesis pathway of GR, were detected. Among them, genes encoding tri-osephosphate isomerase (*POPTR_008G056300*), TPP-dependent pyruvate decarboxylase (*POPTR_016G120100* and *POPTR_011G064000*), galactose mutarotase-like superfamily protein (*POPTR_017G080200* and *POPTR_017G080100*), pyruvate decarboxylase 1 (*POPTR_T102400*, *POPTR_T143400* and *POPTR_004G054100*), alcohol dehydrogenase 1 (*POPTR_002G072100*, *POPTR_T107500* and *POPTR_T107300*) and phosphofructokinase 7 (*POPTR_018G069200*) were significantly upregulated. On the contrary, the gene encoding aldehyde dehydrogenase 2B4 (*POPTR_012G078700*) was dramatically downregulated (Figure 8b).

### 2.6. Secondary Metabolism in Response to Gln Treatment

Tyrosine, phenylalanine and tryptophan are aromatic amino acids (AAAs), which can be used to synthesize protein and as precursors for the synthesis of many substances, such as pigments, isoquinoline alkaloids (IQAs), hormones and cell wall components. In this study, upon exogenous application of Gln, a total of 17 DGEs (including seven upregulated genes and ten downregulated genes) were annotated in tyrosine, phenylalanine and tryptophan metabolism/isoquinoline alkaloid biosynthesis pathway in the GL libraries. In this pathway, ten genes were annotated as polyphenol oxidases (PPO) and were differentially expressed in GL compared with CL. The expression of primary amine oxidase isoform (*POPTR_001G118300* and *POPTR_001G118200*), polyphenol oxidase (*POPTR_011G047300*), tyrosine aminotransferase (*POPTR_017G014200*) and bifunctional aspartate aminotransferase (*POPTR_007G088400*) was significantly upregulated (Table 4, Figure 9a), while the expression of tyrosine decarboxylase (*POPTR_013G052900*) and polyphenol oxidase (*POPTR_001G388900*, *POPTR_T061900*, *POPTR_001G387900*, *POPTR_T062200*, *POPTR_001G388100*, *POPTR_T062100*, *POPTR_011G108300* and *POPTR_001G388400*) was downregulated.

The phenylpropanoid pathway produces an enormous array of secondary metabolites in plants, functioning as structural and signaling molecules that promote plant defense responses against abiotic and biotic stresses. This study detected 66 upregulated and 22 downregulated genes in the phenylpropanoid biosynthesis pathway of GR compared with CR. Among these DEGs, the expression of quinate hydroxycinnamoyl transferase (*POPTR_005G028000*), FAD-binding berberine family protein (*POPTR_001G462800*, *POPTR_001G463100* and *POPTR_011G158700*), peroxidase 64 (*POPTR_005G108900*) and beta-Glucosidase (*POPTR_005G108900*) was significantly suppressed. In contrast, the expression of UDP-glucosyl transferase 73B1 (*POPTR_001G303700*, *POPTR_001G302300* and *POPTR_001G303600*), cinnamoyl CoA reductase (*POPTR_001G045100*), cytochrome P450 CYP73A100-like (*POPTR_001G011000*) and peroxidase 2/3 (*POPTR_016G132900* and *POPTR_012G006800*) was greatly induced (Figure 9b).

### 2.7. Plant Hormone Signal Transduction

Upon exogenous application of Gln, a total of 65 DGEs (including 17 upregulated genes and 48 downregulated genes) and 95 DEGs (including 41 upregulated genes and 54 downregulated genes) were annotated in plant hormone signal transduction in the GL and GR libraries, respectively. Among DEGs in GL, genes encoding the CAP superfamily protein (*POPTR_009G083100*), highly ABA-induced PP2C gene 1 (*POPTR_009G037300*) and probable protein phosphatase 2C 51 (*POPTR_012G002700*) were significantly upregulated, while genes encoding the SAUR-like auxin-responsive protein family (*POPTR_003G113100*), jasmonate-zim-domain protein 10 (*POPTR_001G062500* and *POPTR_003G165000*), protein TIFY 10b (*POPTR_003G068900*) and TIFY domain/divergent CCT motif family protein (*POPTR_006G**139400*) were remarkably downregulated. Among DEGs in GR, genes encoding auxin-responsive GH3 family protein (*POPTR_009G092900*, *POPTR_007G050300* and *POPTR_003G161300*), ethylene response factor 1 (*POPTR_010G072300* and *POPTR_005G223300*), SAUR-like auxin-responsive protein family (*POPTR_005G096400*), CAP superfamily protein (*POPTR_T131500*) and xyloglucan endotransglucosylase/hydrolase (*POPTR_005G007200*) were significantly upregulated, while genes encoding indole-3-acetic acid inducible 18 (*POPTR_006G236200*), phytochrome—associated protein 2 (*POPTR_001G186100*), auxin—responsive GH3 family protein (*POPTR_001G069000*), response regulator 3 (*POPTR_010G037800*) and SAUR—like auxin—responsive protein family (*POPTR_018G063400*) were strongly downregulated (Figure 10).

### 2.8. Validation of RNA-Seq Results

To validate the RNA-Seq results, we randomly selected nine DEGs to perform qRT-PCR analysis. The primer information used in this experiment is shown in Appendix A. The results showed high congruence between the qRT-PCR data and RNA-Seq data (Figure 11), suggesting that the transcriptome data are reliable.

## 3. Discussion

Gln, an N transporter and NH_3_ carrier, plays a vital role in C/N metabolism, cellular homeostasis and biomass accumulation [25]. A few pieces of research have been implemented to study the effect of exogenous Gln on plant regeneration efficiency [32,33,34], biomass accumulation [22,35], biochemical constituents [36,37] and gene expression [27]. Although the widespread effects of Gln on plant growth and development have been disclosed, the underlying mechanism of exogenous Gln in regulating plant growth remains elusive. In this study, we performed an RNA-seq experiment and pairwise comparison analysis of the DEGs in GL and GR versus CL and CR, respectively. GO enrichment analysis of GL vs. CL showed that the DEGs were significantly enriched in intercellular carbohydrate metabolism, intercellular polysaccharide metabolism, cytoplasmic ribosomes, ribosome subunits, extracellular region and structural constituent of ribosome. By contrast, the DEGs in GR vs. CR were significantly enriched in the antibiotic metabolism process, antibiotic catabolism process, cytoplasmic components, cell wall, peroxidase activity and oxidoreductase activity. These results indicate that the above pathways play important roles in poplar ‘Nanlin895’ responding to exogenous Gln-based N fertilizer. The analysis of the KEGG pathway enrichment function of DEGs showed that a large proportion of DEGs in GL was enriched in the ribosome, starch and sucrose metabolism and galactose metabolism. On the contrary, a great number of GR DEGs were enriched in the carbon metabolism, phenylpropanoid biosynthesis and glycolysis/gluconeogenesis, implying that ribosome biosynthesis, sugar metabolism and secondary metabolism might be the principal regulators function in poplar ‘Nanlin895’ to acclimatize to exogenous Gln treatment. Additionally, the GO and KEGG enrichment analysis showed that the DEGs in GL and GR were distinctly enriched in different biological processes, cell components and pathways, suggesting that poplar shoots and roots adopted different mechanisms to accommodate the Gln-N fertilizer.

The balance between protein synthesis and protein degradation determines the growth rate of plants [38]; thus, controlling cell growth inevitably involves the control of ribosome biosynthesis [39]. As an essential regulatory element of organism operation, ribosome synthesis is strictly regulated by the external environment, such as the status of carbon, N and temperature [40,41]. Studies have shown that low energy levels will trigger cells to switch to energy conservation mode to maintain basic functions and viability of cells via inhibiting ribosome synthesis [42,43]. In our study, when Gln was used as a sole N source for poplar, 140 ribosome-related DEGs annotated on ribosomes were found in GL, of which 137 genes were downregulated and 3 genes were upregulated (Figure 6), indicating a negative impact of Gln on protein synthesis. Since ribosome biosynthesis is a process with high energy demand [42,43], it can be speculated that the universal repression of ribosome biosynthesis may be because when fertilized with Gln-N, poplars were inclined to distribute more energy and substances for metabolic processes (such as sugar and amino acids metabolism) other than protein synthesis. This will limit excessive energy input into the ribosome pathway, and resultantly, more energy can be invested to promote the carbon and N metabolism required for growth improvement. In agreement with this hypothesis, it was found that a considerable number of genes related to starch and sugar metabolism, carbon metabolism, carbon fixation and biosynthesis of amino acids (Figure 12) were upregulated in the GL and GR libraries.

Sugar can be used as a crucial source of energy and carbon skeleton in cells and also exert signaling functions. As a critical signaling molecule, sugar modulates many vital physiological processes in photosynthetic plants [44]. Sucrose, the end product of photosynthesis, is the primary sugar transported in plants’ phloem and plays a vital role in sugar, signaling initiation [45]. In our previous study, the concentration of total soluble sugars in poplars fertilized with Gln (0.5 mM) was higher than that of the inorganic N control [25]. In the present study, the expression of the key sucrose synthesis genes (such as *sucrose synthase* and *hexokinase*) was coordinately upregulated with genes (such as *auxin*—*responsive GH3 family protein* and *SAUR*—*like auxin*—*responsive protein*) in the plant hormone signal transduction pathway (Table 2, Figure 7 and Figure 10). These results affirm that sugar metabolism and plant hormone signaling transduction [27] play critical roles in poplar in response to Gln treatment, and it also reveals the involvement of a complex interplay of sugar and hormone signaling in Gln-driven growth.

The growth of plants is constantly challenged by different kinds of environmental conditions. In order to buffer the fluctuation of carbon supply and demand to maintain the growth of heterotrophic organs, plants store and transfer some fixed carbon. In chloroplasts, photosynthesis converts inorganic carbon into organic carbon (including glucose and sucrose), which is of great significance in maintaining carbon content in plant cells [46]. Generally, carbon is stored by starch and transported by sucrose. However, various alternative carbohydrates exist in the plant kingdom, among which RFOs (Raffinose family oligosaccharides) are the most prominent [47]. Glucose can be catalyzed by HK, mutase, guanosine transferase and galactinol synthase (GolS) to generate galactinol, the substrate for RFOs (including raffinose, stachyose and verbascose) biosynthesis. Additionally, sucrose and galactitol can be catalyzed by raffinose synthase (RFS) to produce raffinose and inositol, and raffinose can further be catalyzed by stachyose synthase (STS) to form stachyose, from which a number of different oligosaccharides are generated [47,48]. GolS, as the first enzyme of the RFOs biosynthesis pathway, has a significant effect on RFOs accumulation, consequently regulating the carbon distribution between sucrose and RFOs [49,50]. Studies have shown that under stress conditions (e.g., drought), PSII of the plant was damaged, resulting in a decrease in photosynthetic activity; however, overexpression of *GolS* restored plant photosynthetic activity [51]. In our previous study, the qP and PSII values of poplar under Gln treatment were lower than that of the inorganic N control [25]. The present transcriptome results showed that there were 25 genes enriched in the galactose metabolism pathway in the GL libraries (Table 3), and 6 out of the 13 upregulated genes were annotated as *GolS*. Taken together, it can be proposed that the expression differential of *GolS* between Gln treatment and the inorganic N control is, at least in part, linked to the modulation of photosynthetic performance. Apart from this, the increase in *GolS* expression was in accordance with the upregulation of RFOs synthesis genes, such as *raffinose synthase* and *stachyose synthase*, which are relevant to the increase in plant stress resistance [52]. Moreover, many *GolS* genes in the GL and GR libraries were synergistically upregulated with genes related to secondary metabolism compared with the CL and CR libraries. These results indicate that other than genes associated with growth promotion, external Gln-N also triggered stress response gene-expression.

Tyrosine (Tyr), phenylalanine and tryptophan are aromatic amino acids (AAAs), which are constituted of proteins and precursors for the synthesis of many biomolecules, such as pigments, alkaloids, hormones and cell wall components [53]. These natural products play an important role in plant growth, development, defense and environmental response [53,54]. The α-ketoglutarate and AAAs can generate prephenate by transamination of bifunctional aspartate transaminase (AspAT), and AAAs can produce Tyr by dehydration and transamination which subsequently takes part in multiple metabolic pathways, generating 4-hydroxyphenylcarbonic acid (HP) under the catalyzation of tyrosine transaminase (TYA) or hydroxylating the 3-position of tyrosine to form 3,4-dihydroxy-L-phenylalanine (Dopa) under the catalyzation of polyphenol oxidase (PPOs) [55,56]. In our study, the transcriptome results showed that the expression levels of the aforementioned *AspAT*, *TYA* and *PPOs* genes were significantly changed in Gln-N fertilized poplars compared with the inorganic N control (Figure 9), suggesting that Gln-N activates the biosynthesis of AAAs and AAAs-derived metabolites. It is worth noting that among the 11 *PPOs* annotated in the *P. trichocarpa* genome [55], 9 PPOs were strongly repressed by external Gln-N. PPOs are a group of copper-containing enzymes that catalyze the o-hydroxylation of monophenols to o-diphenols (tyrosinase activity) as well as the oxidation of o-diphenols to quinones (catecholase activity) in the presence of oxygen [57]. As a crucial enzyme of AAAs and isoquinoline alkaloid biosynthesis pathway, PPOs not only function in the formation of pigments, but also play an essential role in the biosynthesis of secondary metabolites [55,56], plant defense against insects and pathogens [58,59,60], and photosynthesis reaction [56,61]. A previous study has demonstrated that the downregulation of *PPOs* activated the Calvin Cycle in poplar [56]. Consistent with this finding, our transcriptome data showed that genes related to Calvin Cycle, such as *phosphoribulokinase* (*PRK*), *triosephosphate isomerase* (*TPI*) and *Glyceraldehyde-3-phosphate dehydrogenase* (*GAPDH*), were remarkably upregulated (Figure 12a). These results suggest that applying Gln might enhance the flow of carbon through the Calvin Cycle, which accelerates the carbon assimilation pathway, consequently improving poplar growth and stress acclimation.

Moreover, the increase in *Fructose-1,6-bisphosphate aldolase* (*FBA*), *GAPDH*, *enolase* and *Pyruvate kinase* (*PK*) and the decrease in *PPOs* also demonstrated activation of glycolysis (Figure 8a), which may provide more ATP for energy metabolism [56,62,63]. This concept is supported by an earlier finding in that the biosynthesis of Gln and glutamate provides an efficient electron pool via dynamic consumption ATP and reduces the energy generated in the process of photosynthesis, so that plants can maintain a relatively stable redox state [64].

## 4. Materials and Methods

### 4.1. Plant Materials and Gln Treatments

In this study, poplar hybrid ‘Nanlin895’ (*Populus deltoides × Populus euramericana*) clones were used as the plant materials unless stated otherwise. The poplar ‘Nanlin895’ seedlings were rooted in 1/2 MS medium for five weeks in the climate chamber under 16 h light/8 h dark at 25 °C/20 °C. The rooted plants were then transferred to water to eliminate endogenous N for two weeks. After that, plants with similar heights were randomly divided into two groups, each of which containing at least 6 plants, and treated with modified Long Ashton (LA) nutrient solution that consisted of 0.5 mM KCl, 0.9 mM CaCl_2_, 0.3 mM MgSO_4_, 0.6 mM KH_2_PO_4_, 42 μM K_2_HPO_4_, 10 μM EDTA·Fe·Na, 2 μM MnSO_4_, 10 μM H_3_BO_3_, 7 μM Na_2_MoO_4_, 0.05 μM CoSO_4_, 0.2 μM ZnSO_4_ and 0.2 μM CuSO_4_ [25], concomitant with either 2 mM inorganic N or 0.5 mM Gln. The hydroponic solution was refreshed every three days. Three biological replicates were performed.

### 4.2. Total RNA Extraction and cDNA Library Construction

Shoots and roots of poplar under 2 mM N and 0.5 mM Gln treatment were harvested after two months and ground into fine powder under liquid N. Total RNA extraction was performed using the RNA Nano 6000 Assay Kit of the Bioanalyzer 2100 system (Agilent Technologies, Santa Clara, CA, USA) and used as input material for the RNA sample preparations. Briefly, mRNA was purified from total RNA using poly-T oligo-attached magnetic beads. Fragmentation was carried out using divalent cations under elevated temperature in the First Strand Synthesis Reaction Buffer (5X). The first strand of cDNA was synthesized using random hexamer primer and M-MuLV Reverse Transcriptase, with the addition of RNaseH to degrade the RNA. The second strand cDNA synthesis was performed using DNA Polymerase I and dNTPs. The remaining overhangs were converted into blunt ends via exonuclease/polymerase activities. After the adenylation of 3′ ends of DNA fragments, an adaptor with a hairpin loop structure was ligated to prepare for hybridization. In order to select cDNA fragments of preferentially 370~420 bp in length, the library fragments were purified with AMPure XP system (Beckman Coulter, Beverly, CA, USA). Then, PCR was performed with Phusion High-Fidelity DNA polymerase, Universal PCR primers and Index (X) Primer. At last, PCR products were purified (AMPure XP system) and library quality was assessed on the Agilent Bioanalyzer 2100 instrument.

### 4.3. Reads Mapping to the Reference Genome

Genome and gene model annotation files of the *Populus trichocarpa* genome were downloaded from the Ensembl and Phytozome websites. The index of the reference genome was built using Hisat2 v2.0.5 and paired-end clean reads were aligned to the reference genome using Hisat2 v2.0.5. The Hisat2 was selected as the mapping tool because it can generate a database of splice junctions based on the gene model annotation file and thus provides a better mapping result than other nonsplice mapping tools.

### 4.4. Differential Expression Gene Analysis

Differential expression analysis of two conditions/groups (three biological replicates per condition) was performed using the DESeq2 R package (1.20.0). DESeq2 provides statistical routines for determining differential expression in digital gene expression data using a model based on the negative binomial distribution. The resulting *p*-values were adjusted using the Benjamini and Hochberg approach for controlling the false discovery rate. Genes with an adjusted *p*-value < 0.05 found by DESeq2 were assigned as differentially expressed.

### 4.5. GO and KEGG Enrichment

Gene Ontology (GO) enrichment analysis of differentially expressed genes was implemented by the clusterProfiler R package, in which gene length bias was corrected. GO terms with corrected *p*-values less than 0.05 were considered significantly enriched by differentially expressed genes. KEGG is a database resource for understanding high-level functions and utilities of the biological system, such as the cell, the organism and the ecosystem, from molecular-level information, especially large-scale molecular datasets generated by genome sequencing and other high-throughput experimental technologies (http://www.genome.jp/kegg/; accessed on 12 August 2022). The clusterProfiler R package was used to test the statistical enrichment of differential expression genes in KEGG pathways.

### 4.6. Quantitative Real-Time Validation of DEGs

Total RNA was extracted from the poplar shoots and roots using an extraction kit (RNAprep Pure Plant Kit, Tiangen, Beijing, China). Reverse transcription was performed using a purified RNA sample by PrimeScript^TM^ RT Reagent Kit (TaKaRa, Dalian, China). Real-time PCR was performed on the StepOnePlus™ Real-Time PCR System (Applied Biosystems, Waltham, MA, USA). Referring to the existing method [65], the following PCR conditions (40 cycles) were used: denaturing for 5 s at 95 °C, annealing for 34 s at 60 °C and extension for 30 s at 72 °C. The primer sequences used for this study are shown in Appendix A.

#### Statistical Analysis

IBM SPSS version 21 software was used for statistical analysis, one-way analysis of variance (ANOVA) was used for data comparison, and Duncan’s test method was used to test the significance level of differences between treatments (*p* < 0.05).

## 5. Conclusions

This study identified a large number of DEGs in poplar ‘Nanlin895’ fertilized with Gln-N compared with the inorganic N. The results indicate that the regulation effect of Gln on poplars involves multiple metabolism processes (including the ribosome biosynthesis, starch and sucrose metabolism, carbon metabolism, carbon fixation, biosynthesis of amino acids, the tyrosine, phenylalanine and tryptophan metabolism pathway, galactose metabolism, and phenylpropanoid pathway) and plant hormone signaling transduction as stated below: (1) Gln inhibited the biosynthesis of the ribosome coinciding with the transcriptional induction of *sucrose synthases*, *sucrose phosphate synthases* and *hexokinase*; therefore, it may limit energy input into protein synthesis and provide more energy for sugar metabolism. (2) A significant co-upregulation of the *GolS* genes, AAAs biosynthesis-related genes and RFOs synthesis genes was found, indicating coordination of the carbon resources redistribution between growth and stress response. (3) Exogenous application of Gln inhibited the expression of *PPOs*-related genes concomitant with the induction of Calvin Cycle-related genes such as *PRK*, *TPI* and *GAPDH*, which may facilitate the flow of carbon toward biomass production while tightly constraining the biosynthesis of secondary metabolites. (4) The downregulation of *PPOs* was in accordance with the increase in glycolysis/gluconeogenesis pathways genes *FBA*, *GAPDH*, *enolase* and *PK*, implying the activation of glycolysis, which was supposed to provide more ATP for energy metabolism. (5) Expression of some plant hormone signaling-associated genes (e.g., auxin signaling pathway) were remarkably altered by exogenous Gln, supporting Gln’s role in signaling transduction. Overall, our RNA-seq results revealed a significant transcriptional distinction between Gln-N (i.e., an amino acid form of N) and the inorganic N in poplars, in which exogenous Gln regulated a more comprehensive range of genes relating to ribosome biogenesis, carbon and N metabolism, and hormone signaling transduction, as a consequence of a practiced energy switch and C/N balance for growth and stress responses. Our study provides updated information to understand the molecular mechanism of different forms of N fertilizers.

## Figures and Tables

**Figure 1 ijms-23-09998-f001:**
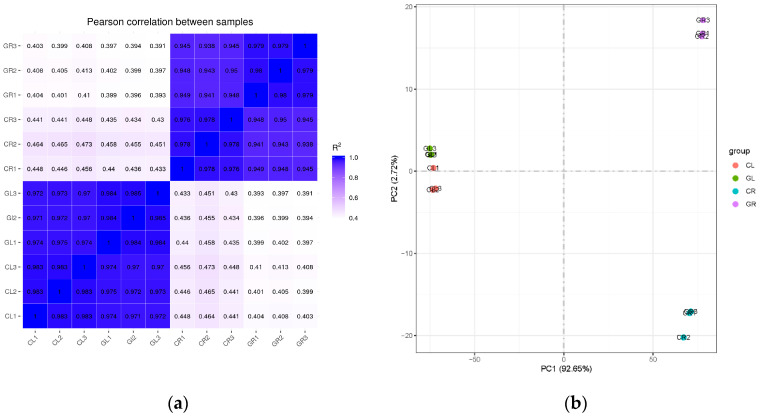
Pearson’s correlation and principal component analysis (PCA). (**a**) Heatmap of correlation matrix showed the correlation coefficient between samples; (**b**) PCA was performed on the biological replicates of each sample set. CL, CR, GL and GR represent the shoots and roots samples of the inorganic N control and the Gln treatment, respectively.

**Figure 2 ijms-23-09998-f002:**
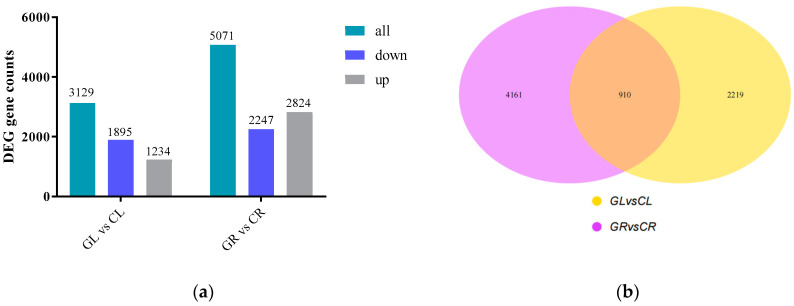
Differentially expressed genes (DEGs) in the shoots (L) and roots (R) of poplars fertilized with Gln (G) or the inorganic N (C). (**a**) The up- and downregulated DEGs by the pairwise comparison of GL vs. CL and GR vs. CR; (**b**) the Venn analysis of DEGs.

**Figure 3 ijms-23-09998-f003:**
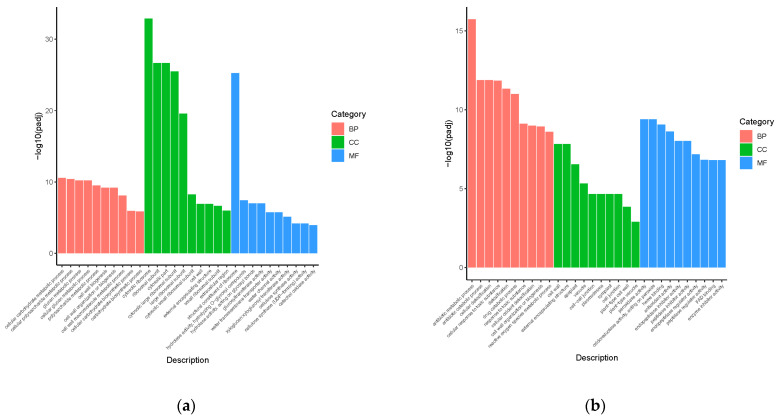
GO classification of differentially expressed genes (DEGs) in the shoots (**a**) and roots (**b**) of poplars fertilized with Gln.

**Figure 4 ijms-23-09998-f004:**
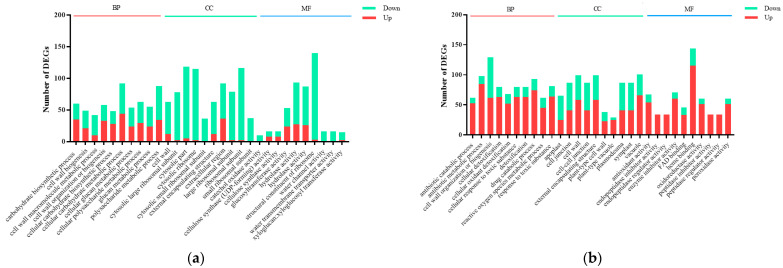
The number of differentially expressed genes (DEGs) in the shoots (**a**) and roots (**b**) of poplars fertilized with Gln compared with the inorganic N control. Red bars indicate numbers of upregulated genes; green bars indicate numbers of downregulated genes.

**Figure 5 ijms-23-09998-f005:**
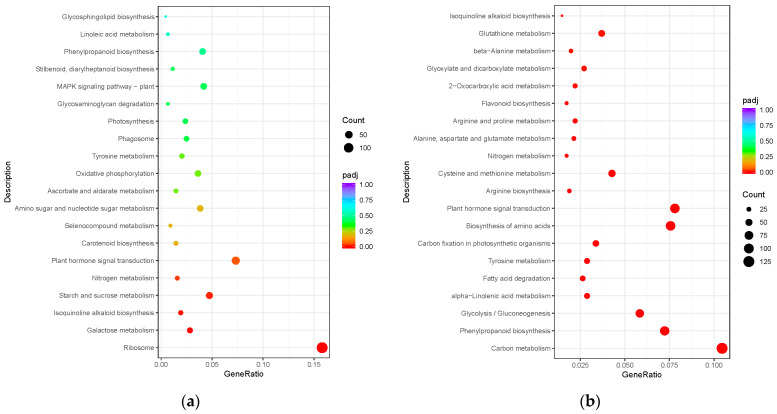
KEGG enrichment of DEGs in the shoots (**a**) and roots (**b**) of poplars fertilized with Gln compared with the inorganic N control. The Y axis corresponds to the KEGG pathway, and the X axis shows the enrichment ratio between the number of DEGs enriched in a particular pathway. The colors of red to green dots correspond to different *padj* ranges, and the sizes of the dots represent numbers of DEGs enriched to the pathway.

**Figure 6 ijms-23-09998-f006:**
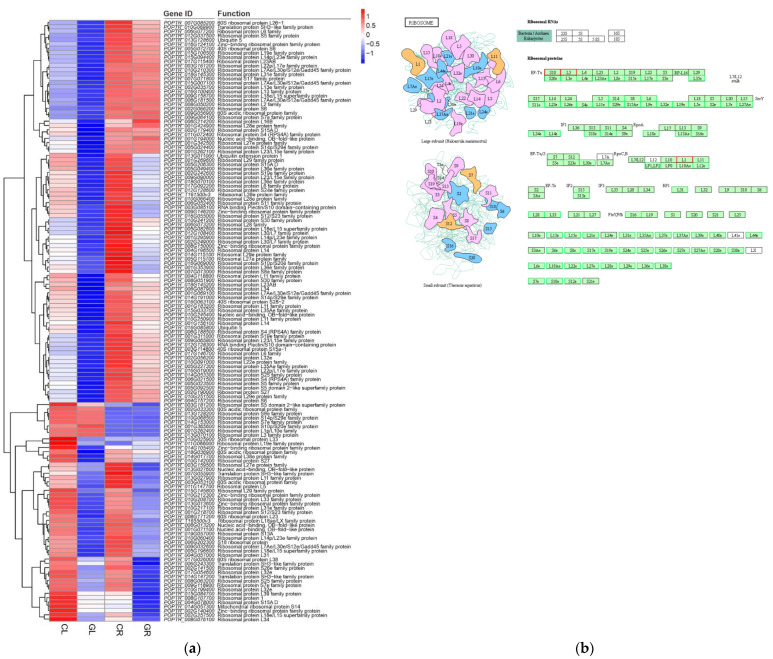
Expression pattern analysis of DEGs in ribosome by the pairwise comparisons of GL vs. CL and GR vs. CR. (**a**) Heatmaps of DEGs in the ribosome pathway; (**b**) KEGG pathway presenting DEGs in ribosome in GL compared with CL. Red boxes indicate upregulation and green boxes indicate downregulation. CL, CR, GL and GR represent the shoots and roots samples of the inorganic N control and the Gln treatment, respectively.

**Figure 7 ijms-23-09998-f007:**
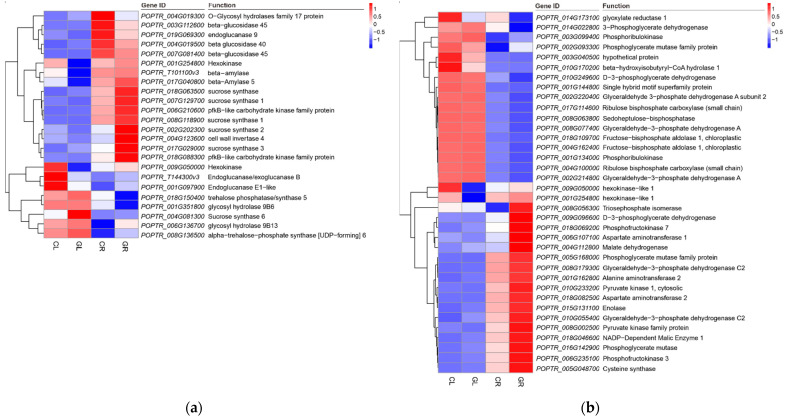
Heatmap of DEGs in starch and sucrose metabolism pathway (**a**) and carbon metabolism (**b**) by the pairwise comparisons of GL vs. CL and GR vs. CR. CL, CR, GL and GR represent the shoots and roots samples of the inorganic N control and the Gln treatment, respectively.

**Figure 8 ijms-23-09998-f008:**
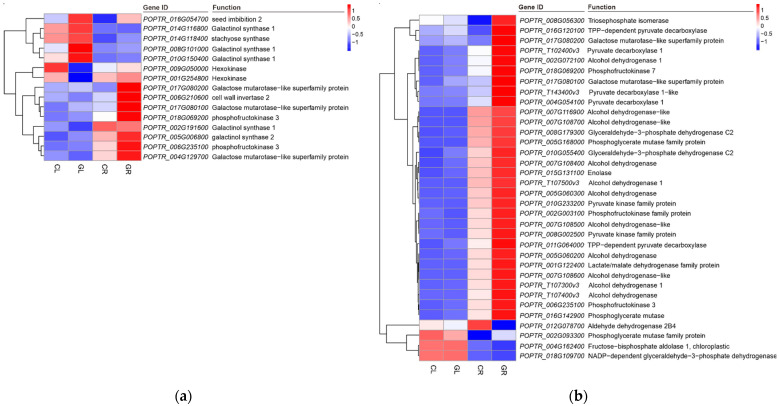
Expression pattern analysis of DEGs in galactose metabolism (**a**) and glycolysis/gluconeogenesis pathway (**b**) by the pairwise comparisons of GL vs. CL and GR vs. CR. CL, CR, GL and GR represent the shoots and roots samples of the inorganic N control and the Gln treatment, respectively.

**Figure 9 ijms-23-09998-f009:**
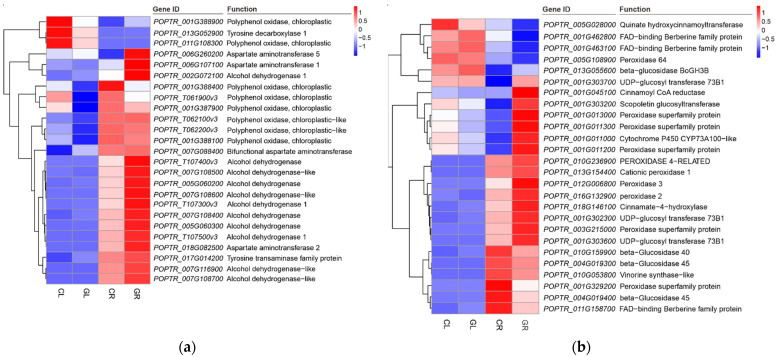
Expression pattern analysis of genes associated with tyrosine, phenylalanine and tryptophan metabolism (**a**) and phenylpropanoid biosynthesis pathway (**b**) by the pairwise comparisons of GL vs. CL and GR vs. CR. CL, CR, GL and GR represent the shoots and roots samples of the inorganic N control and the Gln treatment, respectively.

**Figure 10 ijms-23-09998-f010:**
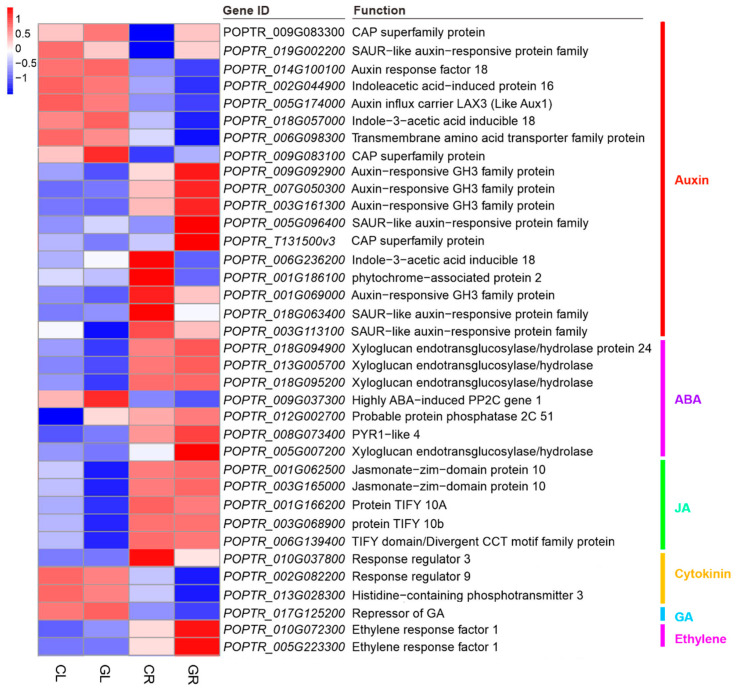
Expression pattern analysis of genes in plant hormone signal transduction by the pairwise comparisons of GL vs. CL and GR vs. CR. CL, CR, GL and GR represent the shoots and roots samples of the inorganic N control and the Gln treatment, respectively.

**Figure 11 ijms-23-09998-f011:**
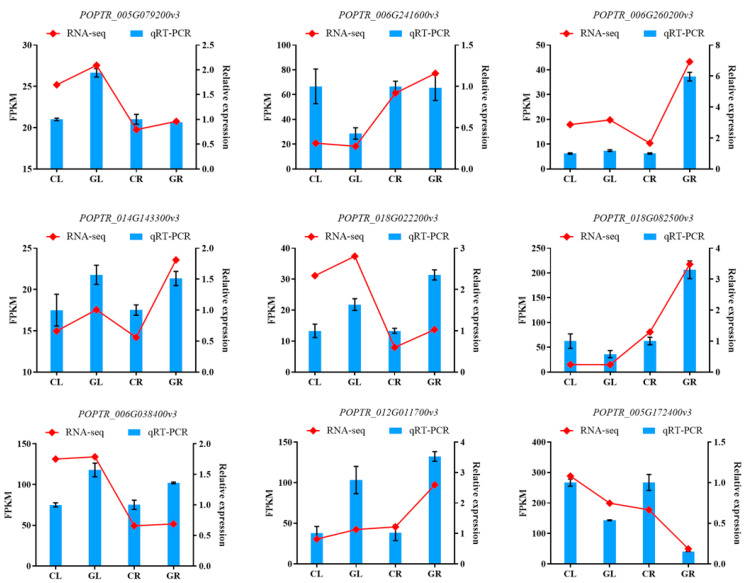
The qRT-PCR validation of selected DEGs in poplars fertilized with Gln compared with the inorganic N. CL, CR, GL and GR represent the shoots and roots samples of the inorganic N control and those of the Gln treatment, respectively.

**Figure 12 ijms-23-09998-f012:**
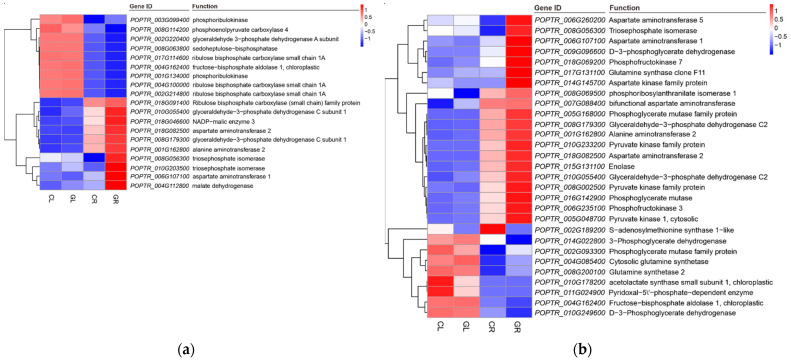
Expression pattern analysis of genes related to carbon fixation (**a**) and amino acids biosynthesis (**b**) by the pairwise comparisons of GL vs. CL and GR vs. CR. CL, CR, GL and GR represent the shoots and roots samples of the inorganic N control and the Gln treatment, respectively.

**Table 1 ijms-23-09998-t001:** Statistics analysis of sequencing data.

Sample	Raw Reads	Clean Reads	Clean Bases	Error Rate (%)	Q30 (%)	GC (%)	Total Mapping (%)	Unique Mapping (%)
CL	44,993,545	43,578,975	6.53G	0.03	93.75	44.29	87.48	85.31
GL	44,483,713	43,239,443	6.48G	0.03	93.83	44.18	88.02	85.77
CR	43,680,084	41,972,709	6.29G	0.03	93.71	43.82	85.51	83.11
GR	43,732,617	41,306,527	6.16G	0.03	93.43	44.20	74.47	72.35

CL, CR, GL and GR represent the shoots and roots samples of the inorganic N control and the Gln treatment, respectively.

**Table 2 ijms-23-09998-t002:** DEGs related to starch and sucrose metabolism in the GL and GR libraries.

Gene	Number of DEGs in CL vs. GL		Number of DEGs in CR vs. GR	
	Upregulated	Sum	Upregulated	Sum
*sucrose synthase*	2	2	3	3
*probable sucrose phosphate synthase*	2	2	1	1
*starch synthase*	2	3	0	1
*granule-bound starch synthase*	2	3	-	-
*probable fructokinase*	-	-	3	3
*alpha-1,4 glucan phosphorylase*	1	1	2	2
*1,4-alpha-glucan-branching enzyme*	2	2	-	-
*4-alpha-glucanotransferase*	-	-	1	2
*endoglucanase*	1	5	2	5
*beta-amylase*	1	2	1	1
*ectonucleotide pyrophosphatase*	-	-	1	1
*hexokinase*	1	3	1	2
*beta-glucosidase*	0	3	3	7
*probable trehalose phosphate phosphatase*	2	4	5	5
*trehalose phosphate synthase*	1	3	2	3
*alpha-amylase*	1	1	2	2
*phosphoglucomutase*	1	1	1	1
*acid beta-fructofuranosidase*	0	2	0	1
*glucose-6-phosphate isomerase*	1	1	1	1
*glucan endo-1,3-beta-glucosidase*	0	1	0	3
*beta-fructofuranosidase*	-	-	1	1

**Table 3 ijms-23-09998-t003:** DEGs related to galactose metabolism in the GL libraries.

Gene	Number of Upregulated	Number of Downregulated	Sum
* galactinol synthase *	6	1	7
* raffinose synthase *	3	2	5
* beta-galactosidase *	1	1	2
* hexokinase *	1	2	3
* alpha-galactosidase *	0	2	2
* stachyose synthase *	1	0	1
* phosphoglucomutase *	1	0	1
* aldose 1-epimerase *	0	1	1
* acid beta-fructofuranosidase *	0	2	2

**Table 4 ijms-23-09998-t004:** DEGs related to tyrosine, phenylalanine and tryptophan metabolism in the GL libraries.

Gene	Number of Upregulated	Number of Dowregulated	Sum
* polyphenol oxidase *	1	9	10
* primary amine oxidase isoform *	2	0	2
* tyrosine aminotransferase *	2	0	2
* bifunctional aspartate aminotransferase *	2	0	2
* tyrosine decarboxylase *	0	1	1

## Data Availability

The raw sequence reads of RNA-seq were deposited in the NCBI with accession BioProject of PRJNA867508 and the accession BioSample, SAMN30202751- SAMN30202762, including twelve accession numbers of SRR20998237-SRR20998248 for triplicate data of Gln treatment (CL, GL, CR and GR).

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
