# Peer review of "Transcriptome Analysis Reveals Critical Genes and Pathways in Carbon Metabolism and Ribosome Biogenesis in Poplar Fertilized with Glutamine"

_ijms, 2022, doi:10.3390/ijms23179998_

Round 1
Reviewer 1 Report
The authors performed transcriptome analyses of both shoots and roots of poplar ‘Nanli895’ fertilized with Gln (G) or the inorganic N (control, C). The results showed that Gln treatment resulted in down-regulation of a large number of ribosomal genes in shoots while most of the DEGs were annotated to carbon metabolism, glycolysis/gluconeogenesis and phenylpropanoid biosynthesis in roots. The results indicated that both carbon and N allocation are account for Gln on poplar growth and photosynthesis impact. This study is informative for researchers to understand the molecular mechanism of different forms of N fertilizers. The following points need to be considered or improved in this manuscript.
1. The images in Figure 3 can not be seen clearly, especially the characters. It’s better use vector map in the manuscript for huge image.
2. In Figure 10, the authors performed clusting of gene expression pattern analysis in plant hormone signaling transduction. However, the results looks confused, it’s better the authors remove the clusting and display the gene expression by hormone name, such as Auxin, ABA, ethylene. Please refer to Urano et al. The Plant Journal (2017) 90, 17–36. doi: 10.1111/tpj.13460.
3. For Populus gene name, the authors used ‘POPTR_013G052900v3, POPTR_005G028000v3…’, the suffix ‘v3’ can be removed in the whole manuscript, writing like POPTR_013G052900 or POPTR_005G028000 is fine.
4. In Line 191, the authors mentioned ‘A heatmap was generated to show the log10 FPKM values of differentially expressed ribosome genes (Figure 6a)’. However, in transcriptome data, some values are 0, how did the authors perform the data transformation?
5. Line 587-588, the sentence ‘including twelve accession numbers of SRRXXXXXXXX-XXX’ is redundant.
Author Response
Response1: Thanks for the comment. The resolution of Figure 3 has been improved.
Response 2: Thanks for the excellent comment. This figure has been updated according to the reviewer’s suggestion. The genes have been regrouped by the corresponding hormone signaling pathways.
Response 3: Thanks for the comment. The suffix ‘v3’ has been removed.
Response 4: Thanks for the comment. Sorry for our carelessness. It is Log10(FPKM+1) instead of log10(FPKM). The sequencing data were processed with the Novogene RNAseq pipeline (Novogene, China). The transcriptome data were transformed according to expanded model matrices. During the data process, different normalization and standardization methods including mean normalization, max-min normalization, non-linear normalization, and Z-score standardization (zero-mean normalization; z=(x-μ)/σ) were performed. Please refer to the following websites:
http://www.bioconductor.org/packages/devel/bioc/vignettes/DESeq2/inst/doc/DESeq2.html#tests-of-log2-fold-change-above-or-below-a-threshold
https://www.statology.org/z-score-normalization/
https://www.ncbi.nlm.nih.gov/pmc/articles/PMC4302049
Response 5: The SRR number has been updated: “including twelve accession numbers of SRR20998237-SRR20998248”.
Reviewer 2 Report
The secret of plant growth was a mystery to mankind for a long time. Ancient Greeks invented organic fertilizers, used by farmers for a long time as the only alternative. The concept of inorganic fertilizer in addition to organic one began to spread at the end of the 18th century. Without nitrogen, there is no growth, and the fact that agricultural yields can be increased by adding inorganic salts to the soil allowed to feed large population in the industrial cities of the 19th century.
With modern methods, this mystery is continually being revealed. It was proved by Formánek and Figala a decade ago that plants uptake of free amino acids is substantial, and not only in nutrient-poor environments as it had been expected before. Not all of their findings were published, but other authors observed similar plant behaviour, as stated in the introduction of this paper.
The paper "Transcriptome Analysis Reveals Critical Genes and Pathways in Carbon Metabolism and Ribosome Biogenesis in Poplar fertilized with Glutamine" by Mei Han, Mingyue Xu, Tao Su, Shizhen Wang, Liangdan Wu, Junhu Feng and Changjun Ding presents a complex answer to very interesting question: what is the difference between organic and inorganic source of nitrogen from the metabolic point of view? Poplar ‘Nanlin895’ was chosen as a model organism, a common tree about which a lot of information is already available from previous research.
The problem is studied by the means of differentially expressed genes and divided into several crucial parts of metabolism: protein biosynthesis, sugar and carbon metabolism, plant hormones and secondary metabolites, including phenylpropanoid pathway which leads to many immunity-related bioactive compounds.
The results indicate possible strengthening of either intercellular carbohydrate metabolism, intercellular polysaccharide metabolism and ribosomes-related biosynthesis, or the antibiotic metabolism process, antibiotic catabolism process, cytoplasmic components and enzyme activity. Thanks to deeper understanding to the mechanism, proper balance of organic and inorganic nitrogen sources could be set to ensure healthy growth of the plant, even under the circumstances where stress factors shift the metabolic balance.
The paper describes a thorough study, it is well organized and given in good, simple scientific English. The topic is very important and it will be beneficial for the readers of IJMS.
However, with that much information, a list of abbreviations would be nice to be added for reader's comfort. Authors can add one within a minor revision. Otherwise, the paper is recommended to be accepted.
Author Response
Response: Thanks for the encouragement and the comments. A short list of abbreviations has been included in the conclusion part.